# Specific Non-Reducing Ends in Heparins from Different Animal Origins: Building Blocks Analysis Using Reductive Amination Tagging by Sulfanilic Acid

**DOI:** 10.3390/molecules25235553

**Published:** 2020-11-26

**Authors:** Pierre A. J. Mourier

**Affiliations:** Sanofi, 13 Quai Jules Guesde, 94403 Vitry sur Seine, France; pierre.mourier@sanofi.com; Tel.: +33-1-5893-8813

**Keywords:** heparin, non-reducing end, heparinase digestion, sulfanilic acid tagging, building blocks quantification

## Abstract

Heparins are linear sulfated polysaccharides widely used as anticoagulant drugs. Their nonreducing-end (NRE) has been little investigated due to challenges in their characterization, but is known to be partly generated by enzymatic cleavage with heparanases, resulting in *N*-sulfated glucosamines at the NRE. Uronic NRE (specifically glucuronic acids) have been isolated from porcine heparin, with GlcA-GlcNS,3S,6S identified as a porcine-specific NRE marker. To further characterize NRE in heparinoids, a building block analysis involving exhaustive heparinase digestion and subsequent reductive amination with sulfanilic acid was performed. This study describes a new method for identifying heparin classical building blocks and novel NRE building blocks using strong anion exchange chromatography on AS11 columns for the assay, and ion-pair liquid chromatography-mass spectrometry for building block identification. Porcine, ovine, and bovine intestine heparins were analyzed. Generally, NRE on these three heparins are highly sulfated moieties, particularly with 3-*O* sulfates, and the observed composition of the NRE is highly dependent on heparin origin. At the highest level of specificity, the isolated marker was only detected in porcine heparin. However, the proportion of glucosamines in the NRE and the proportion of glucuronic/iduronic configurations in the NRE uronic moieties greatly varied between heparin types.

## 1. Introduction

Heparin is a complex heterogeneous linear animal polysaccharide used as a drug for its anticoagulant properties. Heparin is also the starting material for the synthesis of the widely used antithrombotic drug, Low Molecular Weight Heparins (LMWH). Heparins can be extracted from various tissues (lung, intestine or skin) and different animal species (mainly porcine, bovine, and ovine). However, porcine mucosa is currently the only approved starting material for LMWH manufacturing [1,2]. The cost of heparin has continuously increased over the past decade, as a result of increased demand for porcine mucosal heparin (PMH), additional testing to better control quality and contaminant risks [3], pig diseases, and has often been associated with supply shortages. The current African swine fever crisis [4] has dramatically exacerbated these factors, resulting in a virtual doubling of the market price in one year. Maintaining product availability is a serious source of concern for drug agencies such as the U.S Food and Drug Administration (FDA), to the extent that since August 2015, there has been an outreach to manufacturers to diversify supply with heparins prepared from other species and tissues, such as bovine lung and intestinal heparins and ovine intestinal heparin (6th and 7th workshops on the characterization of heparin products, Sao Paulo August 2015 and London 2017). Considering this, heparin identification, differentiation from different species and the detection of adulterated heparin samples is a major concern for heparin manufacturers and drug agencies.

Chemically speaking, heparin is a highly sulfated heterogeneous linear polysaccharide made of disaccharide repeating units comprising an uronic acid (either iduronic or glucuronic) and a glucosamine. Building block analysis, realized either by nuclear magnetic resonance (NMR) spectroscopy [5,6] or by chromatography after heparinase digestion [7,8,9], is still the method of choice to characterize both heparin and LMWH, presenting a rather simple image of a complex material.

For purified samples where quantitative polymerase chain reaction (qPCR) [10] cannot be used, building blocks analysis is used to differentiate the four main types of heparins (bovine lung, bovine intestine, ovine intestine, and porcine intestine) [6,11,12,13]. These methods of differentiation use statistical approaches with multivariate analyses, to different degrees. From a strictly analytical point of view, implementation of these methods is difficult because they require important structural databases on the various existing heparins. Moreover, the use of statistics reflects the failure to identify specific oligosaccharide moieties in every species.

In our laboratory, building block analysis of heparin derivatives after digestion with heparinases I + II + III, has for a long time been performed by strong anion exchange (SAX) chromatography [7] with direct ultraviolet (UV) detection of the Δ4-5 unsaturated oligosaccharides at 232 nm. However, the low stability of the silica-based SAX columns was limiting, resulting in a short lifespan and selectivity of the separation shifting continuously with time. These difficulties became even more acute when the stationary phase traditionally used for this application suffered major manufacturing problems. The extent of these problems meant that it was no longer possible to conduct this type of analysis with the traditional stationary phase [7]. Substitution for an equivalent stationary phase was, therefore, required but this introduced problems associated with variable selectivities between different stationary phases. As we already had a good understanding of reductive amination applied to derivatives of hyaluronan, it was thus decided to use this method to tag the building blocks of heparin obtained by heparinase digestion. The choice of sulfanilic acid was based on its ability to increase building block detectability, and on its capacity to improve chromatographic separation of tagged oligosaccharides by the supplementary sulfate, without excessively modifying its polarity, ensuring that usual desalting and polyacrylamide gel permeation methods could still be used. Heparin building blocks obtained after sulfanilic tagging could be separated by almost all anion exchange techniques used in our laboratory for the separation of sulfated oligosaccharides [14] except for dynamic anion exchange (CTA-SAX) [15], where the salt concentration gradient used for elution cannot break the hydrophobic interactions between the sulfanilic tag and the residual C_18_ bonds of the stationary phase. Interestingly, the Carbopack AS11 SAX columns (Thermo Scientific Dionex, France), typically insufficiently retentive to resolve all disaccharide building blocks, was within effective retention range after reductive amination by sulfanilic acid. Ion pair chromatography [7], compatible with mass spectrometry, also gave excellent results.

Reductive amination is widely used with oligosaccharides [16,17], but in the case of heparinoid digests by heparinase, only one study [9] has obtained interesting results. The aromatic label, 2-aminoacridone, deeply modified solute polarity so that separation had to be performed using reversed phase liquid chromatography, thereby losing the selectivity of anion exchange. In addition, separation was performed on LMWH, where the number of building blocks, already high on heparins, is considerably increased by process fingerprints. The authors identified only some non-reducing end (NRE) building blocks compared to their actual number and did not provide real structural data.

When we started method development, we had extensive experience of ion pair liquid chromatography-mass spectrometry (LC/MS) of heparin digests by heparinase and could systematically detect some NRE building blocks such as sulfated glucosamines and unsaturated disaccharides. LMWHs like enoxaparin [18] or semuloparin/AVE5026 [19] and more specifically short chain fractions, contain oligosaccharide fragments issued from the NRE of the starting heparin (PMH). In exploratory studies, the enoxaparin manufacturing process was applied to synthesize LMWH batches using animal sources other than porcine mucosa. Major differences in NRE residues were observed. Namely, one NRE tetrasaccharide (Mw 1172 Da) was only observed in enoxaparin obtained from PMH. To further explore this observation, the development of an analytical method including NRE building blocks was necessary.

NRE were mostly studied in heparan sulfate [20,21]. However, in heparin, they were apparently not the subject of real investigation. NRE were studied in relation to heparanase and a possible role of heparan NRE in the activation of growth factors. Lindahl et al. first discovered in 1975 [22] that the heparin chain was partially digested in mast cells by an endoglycosidase. This endoglycosidase reduces the initial heparin chain length (60,000–100,000 Da) to the size of commercial heparins (5000–25,000 Da) [23]. Heparanases are involved in tumor progression and angiogenesis [24]. The specificity of heparanase cleavage of heparan was the subject of many studies [25,26]. It appears that the main target of heparanases in the heparan chain is the linkage between glucuronic acids and *N*-sulfo glucosamines giving one N-sulfated glucosamine at the NRE. The NRE of glycosaminoglycans in biological samples were also investigated as biomarkers of mucopolysaccharidoses [27]. In this study, the sample was digested by heparinases, followed by reductive amination with aniline. NRE, mono, di, and trisaccharides were obtained, giving information on the exo-enzyme deficiency.

The aim of our work was thus to define new conditions for heparin building blocks analysis in order to expand the analysis to NRE saccharides, unavailable using the classical method. This aim was facilitated by the reductive amination process that eliminates the anomeric doublet and considerably simplifies the chromatogram. Newly developed and simplified building block analysis and quantification methodologies were tested on oligosaccharides previously only assayed using the classical method, resulting in the identification of novel NRE building blocks. Having demonstrated the efficacy and robustness of these new methodologies, building block analysis and characterization was performed on heparins from bovine intestine, porcine intestine, and ovine intestine.

## 2. Results

### 2.1. Derivatization of Enzymatically Digested Heparin with Sulfanilic Acid

2-picoline borane was chosen as the reducing agent [28] as this non-toxic reagent is easier to use in a pharmaceutical environment than the usual cyanoborohydride. Moreover, the reaction is very effective, while problems of desulfation were observed with cyanoborohydride. Addition of acetic acid is not necessary as the natural acidity of sulfanilic acid proved to be sufficient. The elimination of reagents was performed by desalting on Sephadex G10. The efficiency of this step is improved by sequentially inserting into the 20 mL injection loop 1 mL NaCl 1 M, followed by 2 mL H_2_O, and finally injecting the reduced sample.

Post-reaction, the maximum absorbance wavelength was shifted from 232 nm to 265 nm (Figure 1). The ratio of absorbances at 265 nm for ΔHexUA(2S)-GlcN(NS,6S)-SA (ΔIs^sulf^) and at 232 nm for ΔHexUA(2S)-GlcN(NS,6S) (ΔIs) at pH 3 is 2.5) (structural symbols are listed in Table 1). In the classical method [7], quantification is based on the assumption that the molar extinction coefficients at 232 nm of all Δ4-5 oligosaccharides are identical. After sulfanilic tagging, 265 nm was chosen as the wavelength for quantification, based on the hypothesis that tagged oligosaccharides would have identical molar extinction coefficients. The bases for these hypotheses were stronger in this latter case as the absorption at 265 nm due to the sulfanilic tag is less affected by other UV absorbing moieties (such as acetyl glucosamines or carboxylic acids) than the absorption at 232 nm. Figure 1 shows a comparison of the UV spectrum of ΔIs^sulf^ with that of the NRE disaccharide, IdoA(2S)-GlcN(NS,6S)-SA (Is_id_^sulf^). The Δ4-5 unsaturation increases the absorbance of unsaturated sulfanilic oligosaccharides at 232 nm. A saturated oligosaccharide specific signal can thus be obtained by the suppression of Δ4-5 unsaturated building blocks on the recalculated UV signal 265 nm − 2.21 × 232 nm, as 2.21 corresponds to the ratio of absorbances 265 nm/232 nm for ΔIs^sulf^. In the classical method, oligosaccharides with a *N*-acetylated glucosamine can be easily detected using 200 nm–242 nm signal [7]. After sulfanilic tagging, *N*-acetyl glucosamines have a low, but significant, impact on the UV spectra (Figure 1). It is thus possible to have selective detection using the 200 nm − 1.28 × 265 nm or (200 nm–242 nm) − 0.74 × 265 nm signals.

These ratios (2.21, 1.28, and 0.74) must be adjusted if the pH of the mobile phase is modified, due to its influence on the UV spectra of the building blocks. It should be noted that for ion pair LC/MS, different pH requirements along with the use of UV-absorbing heptylamine (HPTA) and 1,1,1,3,3,3-hexafluoro-2-propanol (HFIP) compounds necessitates the use of different UV ratios.

### 2.2. Chromatography of the Heparin Digests

Chromatograms of a PMH digest by heparinases I + II + III using the AS11 method are shown in Figure 2. Compared to the classical analysis, after reductive amination, new oligosaccharides were detected using the selective 265 nm − 2.21 × 232 nm (**—**) signal corresponding to saturated oligosaccharides. For classic unsaturated building blocks identification is rather obvious, since the selectivity of the initial separation (Figure 2A) remains after the reduction (Figure 2B). The influence of the sulfate of the sulfanilic tag shifts retention times upwards so that ΔHexUA-GlcNAc (ΔIVa), which is not retained initially, is well separated as ΔHexUA-GlcNAc-SA (ΔIVa^sulf^). A schematic depicting heparin building block analysis with sulfanilic tagging can be seen in Figure 3.

Chromatograms of the same sample processed using the ion-pair method are shown in Figure 4. Since UV detection coupled with MS was used, the 265 nm UV signal and the total ion current (TIC) are shown in Figure 4A. MS^E^ ion mode [29] enables easy determination of the number of HPTA adducts and the confirmation of a sulfanilic tag (*m*/*z* 335: GlcNH_2_^sulf^, *m*/*z* 377: GlcNAc^sulf^, *m*/*z* 350: UA^sulf^).

Two reconstructed ion chromatograms (RIC) corresponding to the unsaturated disaccharides (**—**) and tetrasaccharides (**—**) are shown in Figure 4A. They were obtained by adding the *m*/*z* contribution of each component of the mixture. The *m*/*z* values all include the sulfanilic tag corresponding to 157 Da. The NRE building blocks (Figure 4B), comprised *N*-sulfated glucosamines, *N*-sulfated disaccharides and trisaccharides (Figure 5). It must be highlighted that RICs only provide qualitative information, and only the UV absorbance at 265 nm of the corresponding chromatographic peak is reliable for quantitative estimation.

The distribution of these still unidentified NRE building blocks is dependent on heparin source, animal, and organ. Considering the porcine sample analyzed in Figure 2 and Figure 4, at least 50% of the NRE are disaccharides, which begin with uronic acids. The major molecular weight found is 595 Da, corresponding to a saturated disaccharide with three sulfates. In fact, two building blocks at 595 Da are detected and the ratio between these two derivatives is extremely dependent on the type of heparin. The 595 Da molecular weight has been previously identified [8,21] and the structure, IdoA2S-GlcN(NS,6S), was proposed [9,30].

From a strictly chromatographic point of view, the ion-pair method is obviously much more efficient (1.7 µm particle size) than the AS11 method (5 µm particle size). Compared to our previous conditions (hexylamine and pentylamine 15 mM) [14], we observed it is better to lower the concentration of the ion-pairing agent and increase its chain length (HPTA 7.5 mM). The negative ionization mode gives lower number of adducts (here generally less than two) than the positive mode, and the additional acquisition in the MS^E^ mode enables easy calculation of adduct number. The MS acquisition is highly sensitive, and we can detect and identify building blocks present at low concentrations. Different glycoserines, already identified [7], can be detected (6–10 min). They are not tagged with sulfanilic acid unless they display a reducing xylose. Other derivatives (epoxy, trisaccharides, residues of the heparin purification step, hexasaccharides, etc.) are detected, but it is out of the scope of this study to describe them all. Despite its high efficiency, the ion-pair method is probably less selective than AS11. The two disulfated glucosamines (Mw 339^sulf^) are coeluted with ΔHexUA-GlcNS-SA (ΔIVs^sulf^) and ΔHexUA(2S)-GlcNAc-SA (ΔIIIa^sulf^) using the ion-pair method, which is an obvious drawback for quantitation. Similarly, galacturonic building blocks ΔGalA-GlcN(NS,6S)-SA (ΔIIs_gal_^sulf^) and ΔGalA-GlcNS-SA (ΔΙVs_gal_^sulf^) are respectively coeluted with ΔHexUA-GlcN(NS,6S)-SA (ΔIIs^sulf^) and ΔHexUA-GlcNS-SA (ΔIVs^sulf^), whereas they can be separated using the AS11 method. This latter method is considerably advantageous in its simplicity and in the selectivity and stability of the column (the same column has been used extensively for more than 10 years). The transparency of the eluents enables selective detection that simplifies the identification of building blocks. Separation with this method is significantly improved compared to the classical method on Spherisorb SAX [7], namely the improved resolution on AS11 compared with silica-based SAX, the elimination of anomers by the reduction to simplify quantitation, and the sulfanilic tagging to provide supplementary information on NRE building blocks.

### 2.3. Identification of NRE Building Blocks

As mentioned in the introduction, one of the objectives of expanding the heparin building block analysis with sulfanilic tagging was to understand the differences observed in the NRE oligosaccharides of enoxaparin obtained from heparins of various origins. Logically, the proportion of NRE increases when chain length of the fraction decreases.

Initially, we believed that the isolation of the two main NRE tetrasaccharides, present in all LMWH prepared by chemical β-eliminative cleavage, was easier than the 595 Da NRE disaccharides, and was sufficient to enable full chromatographic characterization of the corresponding building blocks. These NRE tetrasaccharides with Mw 1172 Da are easily detected by LC/MS and the proposed structure is usually Is_id_-Is_id_ (IdoA(2S)-GlcN(NS,6S)-IdoA(2S)-GlcN(NS,6S)) [18]. This structure seems logical, knowing that more than 60% of the heparin chain is composed of Is_id_; however, it is in fact only the minor constituent of LMWH synthesized from PMH. The chosen LMWH for the purification was semuloparin [19] (Appendix A). Like enoxaparin, semuloparin is synthesized from PMH by alkaline depolymerization and contains the same NRE tetrasaccharides. However, the purification is easier than with enoxaparin, because its gel permeation chromatogram (GPC) fractions are simpler due to the absence of 1.6-anhydro derivatives and fewer reducing mannose epimers (both enoxaparin alkaline fingerprints). The two isolated structures were IIs_glu_-Is_id_^sulf^ (GlcA-GlcN(NS,3S,6S)-IdoA(2S)-GlcN(NS,6S)-SA) (Appendix A) and Is_id_-Is_id_^sulf^ (Appendix A). It should be noted that the IIs_glu_-Is_id_ structure has already been detected in the enoxaparin tetrasaccharide fraction [31], but by applying NMR to blindly collected samples.

Due to unexpected elution behavior of Is_id_^sulf^ when using the ion pair method, NRE disaccharides required further isolation. The purification, possibly realized from disaccharide fractions of heparin digests by heparinase I alone, was much easier than anticipated. Their structures, expected to be IIs_glu_^sulf^ (GlcA-GlcN(NS,3S,6S)-SA) and Is_id_^sulf^ were confirmed after isolation and full NMR assignment (Appendix A).

IIs_glu_^sulf^ and Is_id_^sulf^ were utilized in the ion pair method, which showed specific retention behavior for Is_id_^sulf^ (Figure 6). The reconstructed chromatograms Mw 595^sulf^ for Is_id_^sulf^ (Figure 6B-1) shows two peaks with identical MS fragmentations. The major peak, coeluted with ΔIs^sulf^, is hardly detectable in digested heparins (Figure 6B-3), as result of signal saturation due to the matrix effect. The minor peak is paradoxically more easily detected (Figure 6B-3) but taking it alone in the quantitation of Is_id_^sulf^, would result in obviously underestimated values. The major benefit of the ion-pair method was therefore its effectiveness at identifying building blocks, but it presented major drawbacks for quantitation. Therefore, the AS11 method was the method of choice for quantitation, but the co-eluting building blocks in Figure 2 required separation.

### 2.4. Treatment by β-Glucuronidase

As NRE in heparins are partially comprised of glucuronic acid, the action of β-glucuronidase which cleaves exolytically glucuronic acids, is an interesting way to explore NRE in heparin. This approach was already used for the study of NRE in heparan [21], detecting hardly explainable GlcA moieties in some samples. 

The influence of the addition of β-glucuronidase to the sulfanilic digest of the porcine heparin used in the AS11 method can be seen in Figure 7. First, it confirms that IIs_glu_^sulf^ is transformed by β-glucuronidase into GlcN(NS,3S,6S)^sulf^, confirming chromatographic identification of the latter. It also appears that the mono-desulfated NRE disaccharides at Mw 515^sulf^ are transformed by β-glucuronidase into GlcN(NS,6S)^sulf^, resulting in the identification of a third NRE disaccharide for PMH, GlcA-GlcN(NS,6S) (IIs_glu_). Ion pair LC/MS of the same sample (Appendix A), detected two different peaks at Mw 515^sulf^, apparently sensitive to β-glucuronidase. During the identification of the two main disaccharides at Mw 595^sulf^, IIs_glu_^sulf^ and Is_id_^sulf^, two other NRE disaccharides at Mw 515^sulf^ were isolated and identified as IIs_glu_^sulf^ and IdoA-GlcN(NS,6S)-SA (IIs_id_^sulf^) (Appendix A). Both were present in PMH digests in almost equal amounts; the presence of IIs_glu_ was confirmed but there is no simple explanation for the presence of IIs_id_.

### 2.5. Treatment by Δ4-5-Glycuronidase

The completion of building blocks identification was achieved with NRE glucosamines by treating the heparin digest with Δ4-5-glycuronidase (Figure 8 for the AS11 method, Appendix A for ion pair).

This enzyme cleaves ΔHexUA moieties. GlcNS^sulf^ and GlcN(NS,6S)^sulf^ are the results of the cleavage of ΔIVs^sulf^ and ΔIIs^sulf^, respectively. GlcN(NS,3S)^sulf^ was also identified by the sulfanilic tagging of GlcNS,3S (commercially available). 

Other building blocks, like ΔIVa^sulf^, ΔHexUA-GlcNAc(6S)-SA (ΔIIa^sulf^), ΔHexUA-GlcNAc(6S)-GlcA-GlcN(NS,3S,6S)-SA (ΔIIa-IIs_glu_^sulf^), ΔHexUA-GlcNAc(6S)-GlcA-GlcN(NS,3S)-SA (ΔIIa-IVs_glu_^sulf^), and ΔHexUA-GlcN(NS,6S)-GlcA-GlcN(NS,3S,6S)-SA (ΔIIs-IIs_glu_^sulf^) were also cleaved by Δ4-5-glycuronidase as expected. Cleavage of ΔIIa^sulf^ gives the elution time of GlcNAc(6S)^sulf^, detected only in low amounts in heparin. 

The trisaccharides obtained by ΔIIa-IIs_glu_^sulf^ and ΔIIs-IIs_glu_^sulf^ cleavages appear to fit with the retention of NRE trisaccharides at Mw 878^sulf^ and Mw 916^sulf^ which could thus correspond to GlcNAc(6S)-GlcA-GlcN(NS,3S,6S)-SA (GlcNAc(6S)-IIs_glu_^sulf^) and GlcNS(NS,6S)-GlcA-GlcN(NS,3S,6S)-SA (GlcN(NS,6S)-IIs_glu_^sulf^). The 3-*O* sulfation of their reducing disaccharide could be one explanation for the resistance of some NRE trisaccharides to heparinase digestion. However, GlcNS-IdoA(2S)-GlcN(NS,6S) (GlcNS-Is_id_) (Mw 836 Da) has already been isolated from digested PMH [32] and the authors underlined the resistance of this trisaccharide to heparinase digestion. Similarly, in another study (unpublished results), we have isolated the trisaccharide GlcNAc-IdoA(2S)-GlcN(NS,6S) (GlcNAc-Is_id_) (Mw 798 Da), from heparinase digested bovine lung heparin (BLH).

### 2.6. Optimization of Chromatographic Conditions for the AS11 Method

Use of the AS11 method to quantify building blocks requires improved separation. An optimization of the chromatographic conditions (Appendix A) was thus conducted. The influence of solvent A (pH 2.5–3.2) has been studied; the retention behaviors appeared to be rather diverse, depending essentially on the presence of uronic carboxylic acids in the building blocks. The modification of the ionic state of the carboxylic acid on the uronic acid generates an increase in retention when the pH is increased in the range around the pKa (≈3) [33]. Logically, the retention times of glucosamine building blocks are almost constant. For disaccharidic building blocks, the single carboxylic moiety gives a positive slope for retention times versus pH (Appendix A). When the building blocks contain two carboxylic acids, i.e., for the 3-*O* sulfated tetrasaccharides, the slopes are steeper than for disaccharides. Thus, it is possible to improve the selectivity between Is_id_^sulf^ and its neighboring building blocks, IIs_glu_^sulf^ and ΔIIa-IVs_glu_^sulf^, but without achieving complete resolution. The latter was obtained by using two columns connected in series. This enabled significant improvement in resolution without any major increase in retention times since the elution gradient is unchanged. Figure 9 shows chromatograms of heparin digests from PMH, ovine (OMH) and bovine (BMH) mucosa heparins using these optimized conditions. Compared to Figure 2, new building blocks such as NRE trisaccharides G(3,4,0) and G(3,5,0) were now detected. GlcNAc(6S), identified by the action of Δ4-5-glycuronidase (Figure 8), was detected in low amounts in our PMH sample. Peaks that eluted at about six minutes (before the peak due to the residual sulfanilic acid, detected at 232 nm) were probably due to glycoserines.

### 2.7. Quantification of Building Blocks

Quantification was performed at 265 nm as this is the maximum of absorbance of building blocks in the UV spectra due to the sulfanilic tag. The *w*/*w* percentage for each component is given by the following formula, similar to that used in the classical method [7].
$$\%\frac{w}{w}=\frac{100\times Mw_{i}\times A_{i}}{\sum_{x}Mw_{x}\times A_{x}}$$

*Mw_i_* and *A_i_* represent the molecular weight and the chromatographic area at 265 nm of the assayed component (*i*), respectively; and *Mw_x_* and *A_x_* the molecular weight and the chromatographic area, respectively, of either the peak *x* or the zone *x* specified by its retention time; the sum being related to all the components eluted. The molecular weights (*Mw*) applied to the building blocks correspond to their status in heparin, that is, as sodium salts but without the sulfanilic tag; these values are reported in the Appendix A, where an example for building blocks quantification is also given (Appendix A).

Glycoserines require special treatment, as they could not be labelled by the reductive amination sulfanilic acid tagging. Therefore, the peak area measured at 232 nm was multiplied by 2.5, the 265 nm/232 nm ratio of sulfanilic response coefficient.

Quantification results, presented as percentage (*w*/*w*) of unsaturated and NRE building blocks, for the three heparin samples (PMH, OMH, and BMH) from Figure 9 are gathered in Table 2. The data for unsaturated building blocks fully support values obtained using the classical method [7] (Appendix A). The overall content of NRE building blocks in Table 2 constitutes 3.5–5% *w*/*w* of the heparin chain, i.e., corresponding to 2–2.5% of the monosaccharide content, depending on the heparin chain length.

## 3. Discussion

The results reported here show that analysis of heparin building blocks can be efficiently performed using heparinase digestion followed by reductive amination sulfanilic acid tagging and chromatography on AS11 columns. Sample preparation is straightforward, chromatographic separation of the components is more efficient due to the presence of a unique representative of each building block (no anomeric pair), and importantly, the greater stability of AS11 columns allows precise and reproducible chromatographic conditions, impossible to obtain on silica based SAX columns due to the continuous shift of the selectivity of these rather unstable stationary phases. 

Quantification of unsaturated building blocks obtained with the new method (Table 2) supports the values obtained using the classical method of analysis (Appendix A) The case of ΔHexUA(2S)-GlcN(NS,3S) (ΔIIIs), already identified in 3-*O* sulfated heparan sulfate [34], illustrates the advantages of the described method. This disaccharide is detected in significant amounts only in BMH, where average levels of 1–1.5% *w/w* (unpublished results) are found. During the classical chromatographic method [7], ΔIIIs is coeluted with ΔIs and assayed in our laboratory with either AS11 or CTA-SAX [15] complementary analyses. After sulfanilic tagging, the selectivity of AS11 columns is maintained and ΔIIIs^sulf^ is still only detected in BMH at 1.3% (Figure 9C). 

This occurrence of untagged building blocks raises the question of the application of the present method to the characterization of LMWH, particularly enoxaparin. In this case, 1.6-anhydro building blocks [9,35,36] like glycoserines, cannot be tagged, and are integrated similarly by applying the 2.5 response coefficient. New peaks not found in heparin samples, corresponding mainly to mannose epimers and unsaturated trisaccharides, are detected and some coelutions sometimes occur. These problems have been solved, but the assay is more complex than with heparin, due to the enoxaparin structural diversity that comes from its numerous alkaline fingerprints.

One of the most relevant results of this study is to define a simple method to enable the structural characterization of NRE in heparins and to highlight differences reflecting the origin of the material analyzed. These differences were first noticed in a study where the enoxaparin manufacturing process was applied to OMH and BMH.

The results of this study may explain some observations made when we applied the enoxaparin manufacturing process to heparins of various animal origins. NRE moieties from BMH and PMH (Table 2) have major differences. However, the LMWH resulting from BMH had so many other structural features incompatible with enoxaparin, than their different NRE residues were not a priority in our study. On the contrary, the LMWH obtained from OMH was structurally similar to enoxaparin, despite differences observed in oligosaccharide chains where residual heparin NRE moieties remained. In ovine derived LMWH-version enoxaparin, odd oligosaccharide NRE glucosamines appeared at almost twice the usual content on GPCs, while, on the contrary, some saturated residues (+18 Da) [18] detected in even GPC fractions of enoxaparin were missing. The increase in odd oligosaccharides in ovine LMWH-version enoxaparins, is the result of the higher percentage of NRE glucosamines in OMH versus PMH (82% vs. 51%, Table 2). The chains displaying the IIs_glu_ moiety on their non-reducing end can also explain the saturated chains missing in ovine LMWH but detected in enoxaparin from PMH (49% vs. 18%).

The higher content of non-reducing GlcN(NS,3S,6S) in OMH versus PMH has been already mentioned in a study based on NMR analyses [11]. The presence of IIs_glu_ at the NRE of PMH has also been observed previously [37]. In this latter work, non-reducing glucuronic acid, possibly linked with GlcN(NS,3S,6S), was detected in NMR analyses of PMH. Similarly, the two major components of the dalteparin (issued from PMH) octasaccharide fraction were identified as GlcA-GlcN(NS,3S,6S)-IdoA(2S)-GlcN(NS,6S)-IdoA(2S)-GlcN(NS,6S)-IdoA(2S)-Mnt6S_2,5anhydr_ (IIs_glu_-Is_id_-Is_id_-IdoA2S-Mnt6S_2,5anhydr_) and IdoA(2S)-GlcN(NS,6S)-IdoA(2S)-GlcN(NS,6S)-IdoA(2S)-GlcN(NS,6S)-IdoA(2S Mnt6S_2,5anhydr_ (Is_id_-Is_id_-Is_id_-IdoA2S-Mnt6S_2,5anhydr_) [38]. Considering our data, it seems likely that these two octasaccharides originate in the heparin NRE.

The occurrence of uronic acids as NRE monosaccharides of heparin chains deserves some discussion. Conversion of macromolecular heparin chains into smaller chains is known to involve heparanase, but among the identified NRE building blocks, only the glucosamines and the trisaccharides (Figure 5) can result from cleavage of the heparin backbone by heparanases. The high 3-*O* sulfation of the NRE glucosamines (≈50% for BMH and OMH, 34% for PMH), also observed on heparan [39,40], supports this specificity of heparanases [25,26].

The presence of NRE uronic acids is more puzzling. They constitute about 50% of the NRE in PMH and BMH but the two cases differ. Specifically, uronic acids in BMH seems to be entirely iduronic, even though the structures of mono-desulfated NRE at 515 Da U(2,2,0) are still partially unknown. As there is no specific structural feature, e.g., 3-*O* sulfation, their structure is largely represented in heparin and they might simply be present as NRE of chains at the end of biosynthesis rather than being generated by post-synthetic enzymatic cleavage. In PMH, NRE uronic acids are constituted by IIs_glu_, the key element of antithrombin III (ATIII)-binding sites, at 40–50%, but found in PMH only at 4–5%. In contrast with iduronic acids, this type of sequence could require specific enzymatic cleavages, though no enzyme has been identified yet.

The comparison of NRE in PMH and OMH indicates that they have equivalent degrees of 3-*O* sulfation. However, in OMH, 3-*O* sulfation is mainly due to GlcN(NS,3S,6S, (78%) while for PMH, the contribution of GlcN(NS,3S,6S) is much lower (30%) but balanced by IIs_glu_ (58%). 

The NRE IIs_glu_ was only identified in PMH, and the presence of this marker was confirmed by ^1^H NMR (Appendix A). Thus, this study provides a new marker for PMH, and an analytical tool to assess its content. It also points out NRE as an interesting sequence to differentiate heparins or LMWH. The case of PMH versus OMH is perhaps more acute, because BMH has such different sulfate distribution that it can be easily detected, even at trace levels. The question of the statistical representativeness of the samples analyzed here should be asked. The presence of IIs_glu_ has been checked using numerous PMH and enoxaparin batches. The same test was performed using all eight OMH samples in our laboratory. Six only contained IIs_glu_ at trace level comparable to the sample analyzed in Table 2. In two of them the content was higher, with molar ratios for IIs_glu_/GlcN(NS,3S,6S) between 0.11 and 0.15, when values between 1.5 and 2 are typically obtained for PMH. These samples were obtained from a different laboratory, with no certainty regarding their traceability; however, a 10% PMH content in these batches offered a potential explanation.

## 4. Materials and Methods

### 4.1. Materials

Semuloparin was supplied by Sanofi (Vitry sur Seine, France). All enzyme lyases from *Flavobacterium heparinum* (Heparinase I (EC 4.2.2.7), Heparinase II (no EC number), Heparinase III (EC 4.2.2.8), and Δ4-5 glycuronidase were obtained from Grampian Enzymes (Aberdeen, Scotland, UK). All other reagents and chemicals were of the highest quality available. β-d-Glucuronidase (from bovine liver type B), sulfanilic acid, and picoline borane were obtained from Sigma-Aldrich (Saint-Quentin-Fallavier, France). d-Glucosamine-2-*N*,3-*O* disulphate disodium salt was obtained from Carbosynth (Berkshire, UK). Water was purified using a Millipore Milli-Q purification system.

### 4.2. Heparin Lyase Digestions

Digestion of heparin (20 µL of a 20 mg/mL solution in water) was performed at room temperature for 48 h in a total volume of 160 µL containing 20 µL of a mixture of the three heparinases (each heparinase is 500 milliunits/mL in a pH 7.0 potassium phosphate buffer [10 mM KH_2_PO_4_ and 0.2 mg/mL of BSA]) and 120 µL of 100 mM sodium acetate buffer (pH 7.0) containing 2 mM Ca(OAc)_2_ and 0.1 mg/mL BSA.

### 4.3. Reductive Amination by Sulfanilic Acid

Oligosaccharides obtained after digestion were diluted to 200 μL with water. They were introduced into an HPLC vial (1.7 mL) containing 4–6 mg of sulfanilic acid and 6–10 mg of picoline borane. The reaction was complete after 8 h at 37 °C. The remaining reagents were removed on Sephadex G10 (column 30 × 2.6 cm) circulated with H_2_O/EtOH, 90/10, *v*/*v*. A 20 mL sample loop was used. The tagged sample diluted in 3 mL H_2_O was injected into the loop after the previous insertion of 1 mL NaCl 1N followed by 2 mL H_2_O. After evaporation of the solution, the digest was diluted in 0.5 mL H_2_O with pH adjustment between 5 and 7 by addition of diluted ammonia.

### 4.4. Analysis by SAX Chromatography on AS11 Columns

In the first part of this study, our conventional AS11 method [14,41] was used. More precisely, the heparin digest was injected (2–5 µL) on an Ionpac AS11 column (25 × 0.21 cm) (ThermoScientific Dionex, Montigny-le-Bretonneux, France) column. The column temperature was set at 40 °C. Mobile phase A was 2.5 mM NaH_2_PO_4_ at pH 2.8, and mobile phase B was an aqueous solution of 2.5 mM NaH_2_PO_4_ with 1 M NaClO_4_ adjusted to pH 3.0. A linear gradient (t0 min B% 0; t 80 min B% 60) was applied for elution at a flow rate of 0.22 mL/min. Diode array detection was used. Double UV detection was performed at 265 nm and 232 nm. An NRE building block-specific signal was obtained by the reconstruction of 265 nm − 2.21 × 232 nm. 

In the second part of the study, full resolution of building blocks was obtained with optimized pH for the mobile phase and connection of two AS11 columns (25 × 0.21 cm) in series. Mobile phase A was 2.5 mM NaH_2_PO_4_ at pH 3.2. All other parameters were unchanged. 

### 4.5. Ion Pair LC/MS

Heparin digests were injected on ion-pair LC/MS chromatography using Acquity UPLC BEH C18 column, 2.1 × 150 mm, 1.7 μm (Waters). Mobile phase A was water, and mobile phase B water/acetonitrile (30:70). The ion pairing reagent, HPTA (7.5 mM) and a buffering agent, HFIP (50 mM) were added to both A and B. A linear gradient (t0 min B% 1; t 60 min B% 60) was applied for elution at a flow rate of 0.22 mL/min. Column temperature was set at 30 °C and diode array detection used. Double UV detection was performed at 265 nm and 232 nm. An NRE building block-specific signal was obtained by the reconstruction of 265 nm − 2.6 × 232 nm.

Electrospray ionization (ESI) mass spectra were obtained using a Waters Xevo Q-Tof mass spectrometer. The electrospray interface was set in negative ion mode with a capillary voltage of 2000 V and a sampling cone voltage of 20 V. The source and the desolvation temperatures were 120 °C and 300 °C, respectively. Nitrogen was used as desolvation (750 L/min) and cone gas (25 L/min). The mass range was 50–2500 Da (scan rate = 0.8 s). Acquisition was performed in MS^E^ mode [29] with low energy at 7 V and a high energy ramp from 30 V to 50 V.

### 4.6. Identification of NRE Tetra and Disaccharides

Semuloparin was injected (2 g per injection) on a column (200 × 5 cm) packed with Bio Gel P30. The tetrasaccharide fraction was collected and desalted on a column packed with Sephadex G10 (100 × 7 cm). The reductive amination of the tetrasaccharide batch was a slightly modified version of the procedure used for digests. In 1 mL H_2_O, 50–80 mg of the fraction was added to 50 mg of sulfanilic acid mixed with 50 mg of picoline borane. After 8 h at 37 °C, the reagents were eliminated using Sephadex G10. Semi-preparative purification was performed on AS11 columns (25 × 2.1 cm) at room temperature and a flow rate of 20 mL/min. An aqueous NaClO_4_ (0–0.6 M) mobile phase at pH 2.5 was used for elution. Fractions were neutralized and controlled on analytical AS11. The selection of fractions was based on the NRE selective UV signal.

In a second step, the two main NRE disaccharides were also isolated to check their retention behavior in both ion-pair and AS11 methods. They were isolated from disaccharide fractions of BLH and PMH heparins digested by heparinase 1, purified by conventional methods, as already described [42]. The semi preparative purification was performed using the same methodology as for NRE tetrasaccharides.

### 4.7. Reaction with Exoglycosidases: β-*d*-Glucuronidase

Sulfanilic tagged heparin digests and semuloparin tetrasaccharides were treated with β-d-glucuronidase. Briefly, 100 µL of the oligosaccharide solution was diluted 1/3 in 50 mM sodium acetate pH 4.5, added with 100 units of enzyme and incubated overnight at 30 °C.

### 4.8. Reaction with Δ4-5 Glycuronidase

The sulfanilic tagged heparin digests (100 µL) were diluted 1/2 in 5 mM Na_2_HPO_4_ pH 7 before being treated for one day at room temperature with Δ4-5 glycuronidase (20 milliunits).

## 5. Conclusions

Derivatization by reductive amination sulfanilic acid tagging was applied to heparinase digests of heparin. Two chromatographic methods were proposed for the separation of the tagged building blocks: SAX chromatography on AS11 columns with UV detection and ion-pair liquid chromatography on a C18 column with MS detection. In addition to known heparin classical di and tetrasaccharides, new building blocks issued from the NRE are described. The latter include sulfated glucosamines, uronic acids, and trisaccharides. NRE glucuronic acids were specifically detected in PMH. A marker of porcine heparin, GlcA-GlcNS,3S,6S was isolated and heparin building blocks assayed using SAX chromatography on AS11 columns. The stability of this stationary phase enabled good control of chromatographic separation, which is superior compared to the classical method of separating Δ4-5 unsaturated building blocks on a silica based SAX column [7]. This method was applied to PMH, OMH and BMH heparins. Data obtained for classical building blocks were compatible with that obtained using the classical method. Furthermore, the method gives access to the content of NRE building blocks identified in heparins and points out substantial differences between NRE from heparins of different animal sources. In conclusion, the method described provides a new analytical tool for differentiation of heparin of various animal origins.

## Figures and Tables

**Figure 1 molecules-25-05553-f001:**
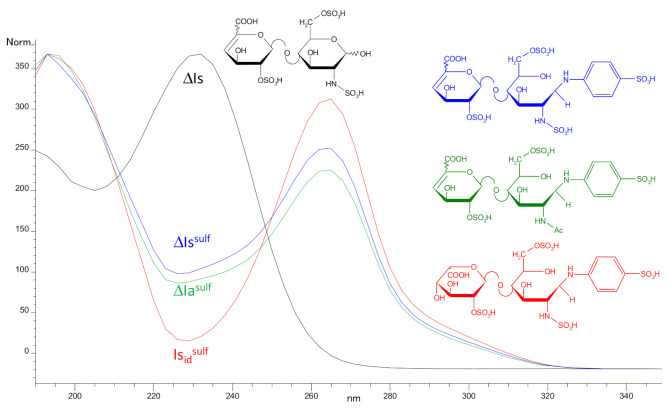
Ultraviolet (UV) spectrum of building blocks (**—** ΔIs, **—** ΔIs^sulf^, **—** ΔIa^sulf^ (ΔHexUA(2S)-GlcNAc(6S)-SA), **—** Is_id_^sulf^).

**Figure 2 molecules-25-05553-f002:**
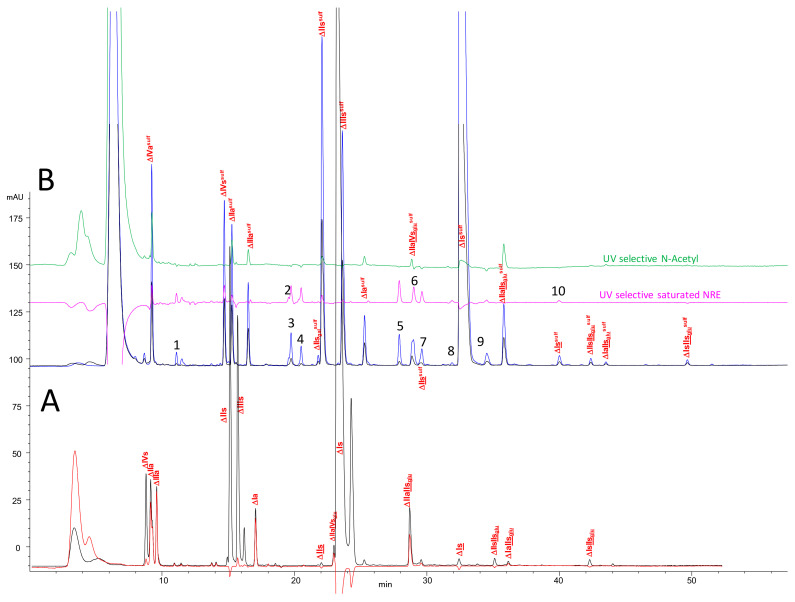
Chromatogram on AS11 of a digest of PMH. (**a**) without reductive amination (**b**) with sulfanilic tagging. Detection:**—**265 nm; **―** 232 nm; **—** 202–242 nm; **—** 265 nm − 2.21 × 232 nm; **—**200 nm − 1.28 × 265 nm. Peak assignment: 1: GlcNS^sulf^, 2: GlcN(NS,3S)^sulf^, 3: GlcN(NS,6S)^sulf^, 4: Mw 515^sulf^ U(2,1,0) ^sulf^, 5: IIs_glu_^sulf^: (GlcA-GlcN(NS,3S,6S)-SA), 6: Is_id_^sulf^: IdoA(2S)-GlcN(NS,6S)-SA, 7: GlcN(NS,3S,6S)^sulf^, 8: Mw 836^sulf^ G(3,4,0) ^sulf^, 10: Mw 916^sulf^ G(3,5,0) ^sulf^, ΔIVa: ΔHexUA-GlcNAc, ΔIVs: ΔHexUA-GlcNS, ΔIIa: ΔHexUA-GlcNAc(6S), ΔIIIa: ΔHexUA(2S)-GlcNAc, ΔIIs_gal_: ΔGalA-GlcN(NS,6S), ΔIIs: ΔHexUA-GlcN(NS,6S), ΔIIIs: ΔHexUA(2S)-GlcN(NS), ΔIa: ΔHexUA(2S)-GlcNAc(6S), ΔIIs = ΔHexUA-GlcN(NS,3S,6S), ΔIIa-IVs_glu_: ΔHexUA-GlcNAc(6S)-GlcA-GlcN(NS,3S), ΔIs: ΔHexUA(2S)-GlcN(NS,6S), ΔIIa-IIs_glu_: ΔHexUA-GlcNAc(6S)-GlcA-GlcN(NS,3S,6S), ΔIs: ΔHexUA(2S)-GlcN(NS,3S,6S), ΔIIs-IIs_glu_: ΔHexUA-GlcNS(NS,6S)-GlcA-GlcN(NS,3S,6S), ΔIa-IIs_glu_: ΔHexUA(2S)-GlcNAc(6S)-GlcA-GlcN(NS,3S,6S), ΔIs-IIs_glu_: ΔHexUA(2S)-GlcNS(NS,6S)-GlcA-GlcN(NS,3S,6S)

**Figure 3 molecules-25-05553-f003:**
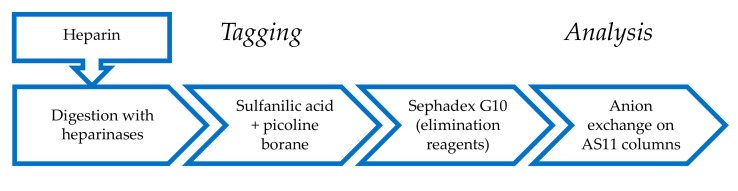
Building block analysis of a heparin sample using sulfanilic tagging.

**Figure 4 molecules-25-05553-f004:**
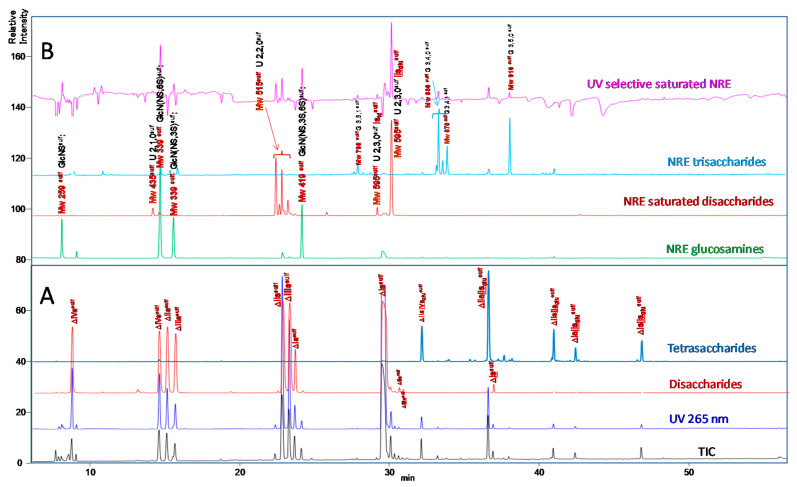
Chromatogram on ion-pair LC/MS of a digest of PMH with sulfanilic tagging. (**A**) Disaccharides: **—** RIC *m*/*z* 535.1 + 573.1 + 615.1 + 653 + 695 + 848.1 + 1043.3, Tetrasaccharides: **—** RIC *m*/*z* 555.6 + 595.5 + 614.5 + 635.5 + 712.1 + 565.5 + 574.5. (**B**) NRE Glucosamines: **—***m*/*z* 415.1 + 495 + 575, NRE saturated disaccharides: **—***m*/*z* 591.1 + 671.1 + 751 + 1061.3, NRE trisaccharides: **— ***m*/*z* 455.5 + 495.5 + 535.5 + 633 + 476.6, **—**Saturated oligosaccharides selective signal: 265 nm − 2.7 × 232 nm; Peak assignment: IIs_glu_^sulf^: GlcA-GlcN(NS,3S,6S)-SA, Is_id_^sulf^: IdoA(2S)-GlcN(NS,6S)-SA, ΔIVa^sulf^: ΔHexUA-GlcNAc-SA, ΔIVs^sulf^: ΔHexUA-GlcNS-SA, ΔIIa^sulf^: ΔHexUA-GlcNAc(6S)-SA, ΔIIIa^sulf^: ΔHexUA(2S)-GlcNAc-SA, ΔIIs_gal_^sulf^: ΔGalA-GlcN(NS,6S)-SA, ΔIIs^sulf^: ΔHexUA-GlcN(NS,6S)-SA, ΔIIIs^sulf^: ΔHexUA(2S)-GlcN(NS)-SA, ΔIa^sulf^: ΔHexUA(2S)-GlcNAc(6S)-SA, ΔIIs^sulf^: ΔHexUA-GlcN(NS,3S,6S)-SA, ΔIIIs^sulf^: ΔHexUA(2S)-GlcN(NS,3S)-SA, ΔIIa-IVs_glu_^sulf^: ΔHexUA-GlcNAc(6S)-GlcA-GlcN(NS,3S)-SA, ΔIs^sulf^: ΔHexUA(2S)-GlcN(NS,6S)-SA, ΔIIa-IIs_glu_^sulf^: ΔHexUA-GlcNAc(6S)-GlcA-GlcN(NS,3S,6S)-SA, ΔIs^sulf^: ΔHexUA(2S)-GlcN(NS,3S,6S)-SA, ΔIIs-IIs_glu_^sulf^: ΔHexUA-GlcNS(NS,6S)-GlcA-GlcN(NS,3S,6S)-SA, ΔIa-IIs_glu_^sulf^: ΔHexUA(2S)-GlcNAc(6S)-GlcA-GlcN(NS,3S,6S)-SA, ΔIs-IIs_glu_^sulf^: ΔHexUA(2S)-GlcNS(NS,6S)-GlcA-GlcN(NS,3S,6S)-SA.

**Figure 5 molecules-25-05553-f005:**
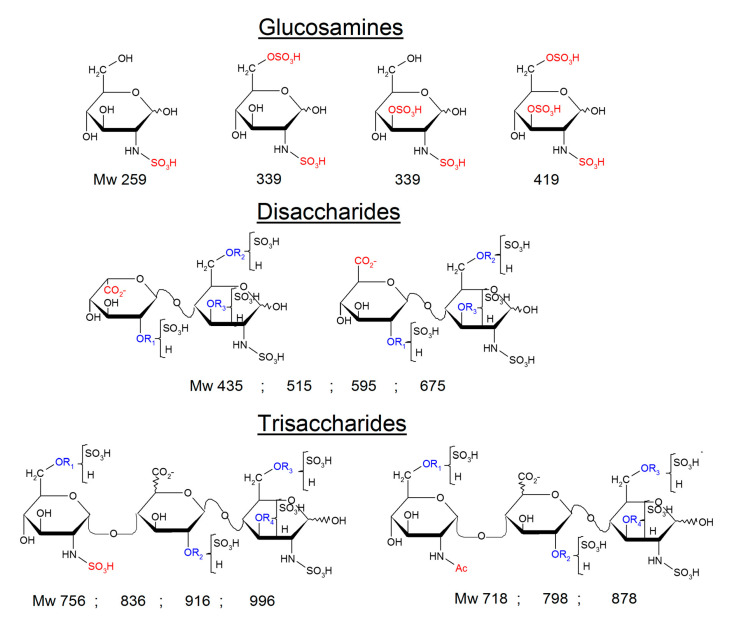
Building blocks corresponding to nonreducing-end (NRE) detected in liquid chromatography-mass spectrometry (LC/MS) of heparin digests.

**Figure 6 molecules-25-05553-f006:**
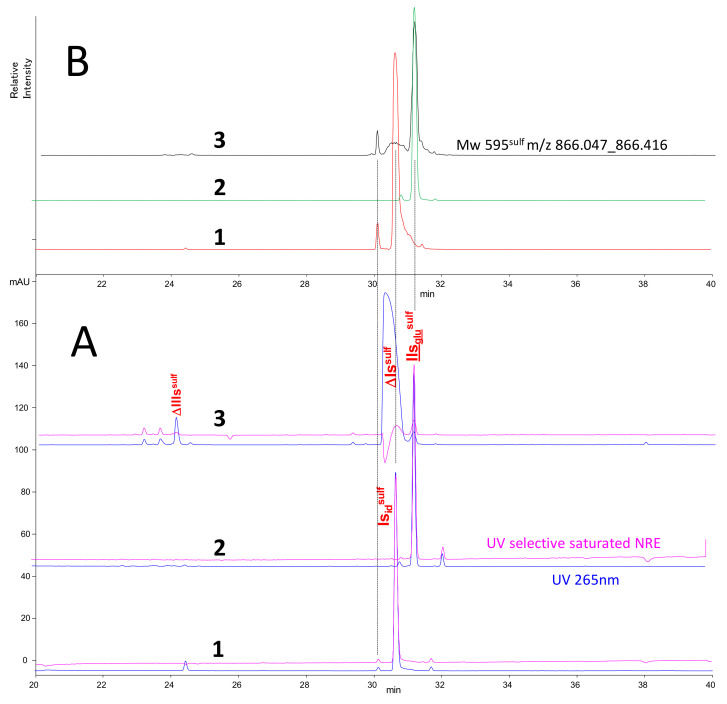
Chromatograms in ion pair chromatography of disaccharide NRE building blocks Is_id_^sulf^ (1), IIs_glu_^sulf^ (2) compared to a sulfanilic tagged disaccharide fraction of porcine heparin digested by heparinase 1 (3). Detection: (**A**) UV**—**265 nm; **—**265 nm − 2.6 × 232 nm; (**B**) RIC Mw 595^sulf^
*m*/*z* 866.2, Peak assignment: IIs_glu_^sulf^: GlcA-GlcN(NS,3S,6S)-SA, Is_id_^sulf^: IdoA(2S)-GlcN(NS,6S)-SA, ΔIIIs^sulf^: ΔHexUA(2S)-GlcN(NS)-SA, ΔIs^sulf^: ΔHexUA(2S)-GlcN(NS,6S)-SA

**Figure 7 molecules-25-05553-f007:**
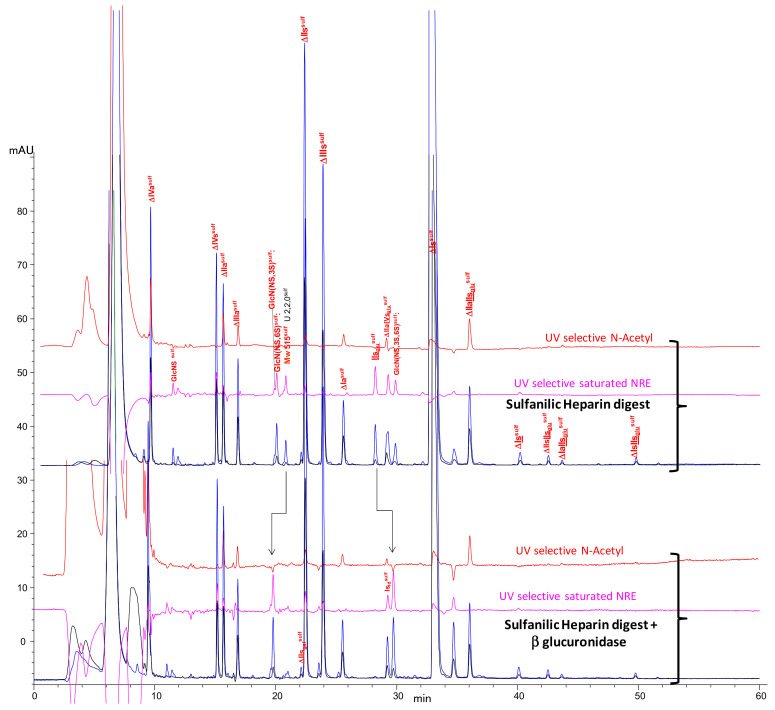
Influence on the AS11 chromatogram of β-d-glucuronidase addition to the heparinase digest of PMH with sulfanilic tagging. Detection:** — **265 nm; **—** 232 nm; **— **200 nm − 1.28 × 265 nm; **—**265 nm − 2.2 × 232 nm, Peak assignment: IIs_glu_^sulf^: GlcA-GlcN(NS,3S,6S)-SA, Is_id_^sulf^: IdoA(2S)-GlcN(NS,6S)-SA, ΔIVa^sulf^: ΔHexUA-GlcNAc-SA, ΔIVs^sulf^: ΔHexUA-GlcNS-SA, ΔIIa^sulf^: ΔHexUA-GlcNAc(6S)-SA, ΔIIIa^sulf^: ΔHexUA(2S)-GlcNAc-SA, ΔIIs_gal_^sulf^: ΔGalA-GlcN(NS,6S)-SA, ΔIIs^sulf^: ΔHexUA-GlcN(NS,6S)-SA, ΔIIIs^sulf^: ΔHexUA(2S)-GlcN(NS)-SA, ΔIa^sulf^: ΔHexUA(2S)-GlcNAc(6S)-SA, ΔIIs^sulf^: ΔHexUA-GlcN(NS,3S,6S)-SA, ΔIIIs^sulf^: ΔHexUA(2S)-GlcN(NS,3S)-SA, ΔIIa-IVs_glu_^sulf^: ΔHexUA-GlcNAc(6S)-GlcA-GlcN(NS,3S)-SA, ΔIs^sulf^: ΔHexUA(2S)-GlcN(NS,6S)-SA, ΔIIa-IIs_glu_^sulf^: ΔHexUA-GlcNAc(6S)-GlcA-GlcN(NS,3S,6S)-SA, ΔIs^sulf^: ΔHexUA(2S)-GlcN(NS,3S,6S)-SA, ΔIIs-IIs_glu_^sulf^: ΔHexUA-GlcNS(NS,6S)-GlcA-GlcN(NS,3S,6S)-SA, ΔIa-IIs_glu_^sulf^: ΔHexUA(2S)-GlcNAc(6S)-GlcA-GlcN(NS,3S,6S)-SA, ΔIs-IIs_glu_^sulf^: ΔHexUA(2S)-GlcNS(NS,6S)-GlcA-GlcN(NS,3S,6S)-SA

**Figure 8 molecules-25-05553-f008:**
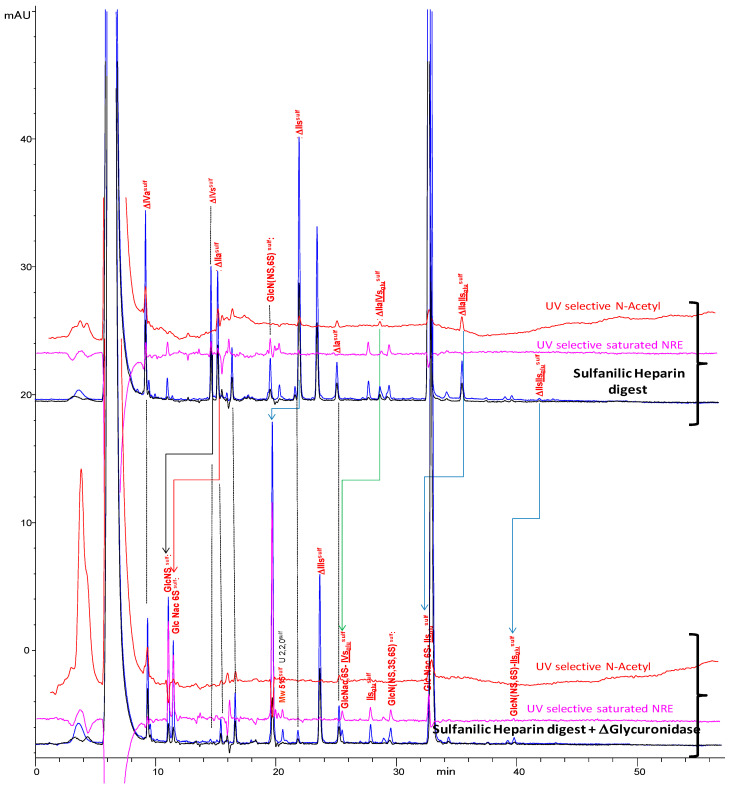
Influence on the AS11 chromatogram of the addition of Δ4-5-glycuronidase of the heparinase sulfanilic digest of heparin. Detection: **—** 232 nm; **—** 265 nm; **— **200 nm − 1.28 × 265 nm; **—**265 nm − 2.2 × 232 nm Peak assignment: IIs_glu_^sulf^: GlcA-GlcN(NS,3S,6S)-SA, Is_id_^sulf^: IdoA(2S)-GlcN(NS,6S)-SA, ΔIVa^sulf^: ΔHexUA-GlcNAc-SA, ΔIVs^sulf^: ΔHexUA-GlcNS-SA, ΔIIa^sulf^: ΔHexUA-GlcNAc(6S)-SA, ΔIIIa^sulf^: ΔHexUA(2S)-GlcNAc-SA, ΔIIs^sulf^: ΔHexUA-GlcN(NS,6S)-SA, ΔIIIs^sulf^: ΔHexUA(2S)-GlcN(NS)-SA, ΔIa^sulf^: ΔHexUA(2S)-GlcNAc(6S)-SA, ΔIIs^sulf^: ΔHexUA-GlcN(NS,3S,6S)-SA, ΔIIIs^sulf^: ΔHexUA(2S)-GlcN(NS,3S)-SA, ΔIIa-IVs_glu_^sulf^: ΔHexUA-GlcNAc(6S)-GlcA-GlcN(NS,3S)-SA, ΔIs^sulf^: ΔHexUA(2S)-GlcN(NS,6S)-SA, ΔIIa-IIs_glu_^sulf^: ΔHexUA-GlcNAc(6S)-GlcA-GlcN(NS,3S,6S)-SA, ΔIs^sulf^: ΔHexUA(2S)-GlcN(NS,3S,6S)-SA, ΔIIs-IIs_glu_^sulf^: ΔHexUA-GlcNS(NS,6S)-GlcA-GlcN(NS,3S,6S)-SA.

**Figure 9 molecules-25-05553-f009:**
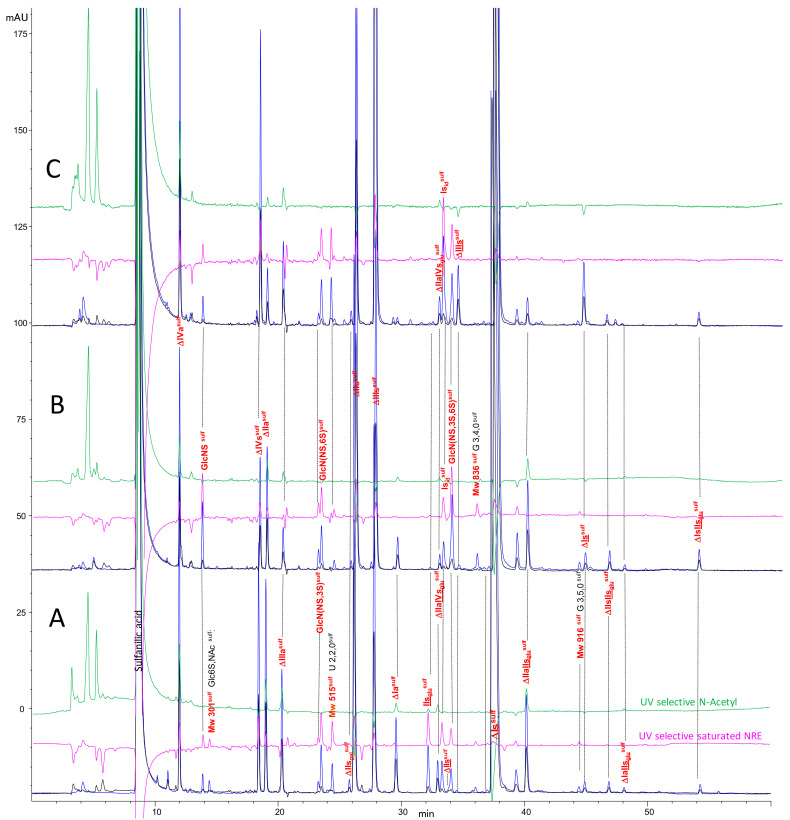
Chromatogram on AS11 (two columns) of heparin digests (**A**) PMH, (**B**) OMH, (**C**) BMH with sulfanilic tagging. Detection: **—** 265 nm; **—** 232 nm; **—** 265 nm – 2.2 × 232 nm; **—** (200–242 nm) − 0.74 × 265 nm Peak assignment: IIs_glu_^sulf^: GlcA-GlcN(NS,3S,6S)-SA, Is_id_^sulf^: IdoA(2S)-GlcN(NS,6S)-SA, ΔIVa^sulf^: ΔHexUA-GlcNAc-SA, ΔIVs^sulf^: ΔHexUA-GlcNS-SA, ΔIIa^sulf^: ΔHexUA-GlcNAc(6S)-SA, ΔIIIa^sulf^: ΔHexUA(2S)-GlcNAc-SA, ΔIIs_gal_^sulf^: ΔGalA-GlcN(NS,6S)-SA, ΔIIs^sulf^: ΔHexUA-GlcN(NS,6S)-SA, ΔIIIs^sulf^: ΔHexUA(2S)-GlcN(NS)-SA, ΔIa^sulf^: ΔHexUA(2S)-GlcNAc(6S)-SA, ΔIIs^sulf^: ΔHexUA-GlcN(NS,3S,6S)-SA, ΔIIIs^sulf^: ΔHexUA(2S)-GlcN(NS,3S)-SA, ΔIIa-IVs_glu_^sulf^: ΔHexUA-GlcNAc(6S)-GlcA-GlcN(NS,3S)-SA, ΔIs^sulf^: ΔHexUA(2S)-GlcN(NS,6S)-SA, ΔIIa-IIs_glu_^sulf^: ΔHexUA-GlcNAc(6S)-GlcA-GlcN(NS,3S,6S)-SA, ΔIs^sulf^: ΔHexUA(2S)-GlcN(NS,3S,6S)-SA, ΔIIs-IIs_glu_^sulf^: ΔHexUA-GlcNS(NS,6S)-GlcA-GlcN(NS,3S,6S)-SA, ΔIa-IIs_glu_^sulf^: ΔHexUA(2S)-GlcNAc(6S)-GlcA-GlcN(NS,3S,6S)-SA, ΔIs-IIs_glu_^sulf^: ΔHexUA(2S)-GlcNS(NS,6S)-GlcA-GlcN(NS,3S,6S)-SA.

**Table 1 molecules-25-05553-t001:** Nomenclature and structural symbols.

**Nomenclature**	
HexUA = Uronic acid	IdoA = l-iduronic acid
GlcA = d-glucuronic acid	ΔHexUA = 4,5-unsaturated uronic acid
GlcN = d-glucosamine	Man = d-mannosamine
NS = *N*-sulfate	NAc = *N*-acetyl
Mnt 6S_2,5 anhydr_ = 6-*O* sulfated 2,5 anhydro Mannitol	2S = 2-*O*-sulfate
3S = 3-*O*-sulfate	6S = 6-*O*-sulfate
GalA = d-galacturonic acid	Epoxy = Epoxised iduronic acid
SA = tagging with sulfanilic acid	*w*/*w* = weight/weight
PMH = Porcine mucosa heparin	BMH = Bovine mucosa heparin
OMH = Ovine mucosa heparin	
**Structural Symbols**	
ΔIVa = ΔHexUA − GlcNAc	ΔIVs = ΔHexUA − GlcNS
ΔIIa = ΔHexUA − GlcNAc(6S)	ΔIIIa = ΔHexUA(2S) − GlcNAc
ΔIIs = ΔHexUA − GlcN(NS,6S)	ΔIIIs = ΔHexUA(2S) − GlcN(NS)
ΔIa = ΔHexUA(2S) − GlcNAc(6S)	ΔIs = ΔHexUA(2S) − GlcN(NS,6S)
ΔIIs = ΔHexUA − GlcN(NS,3S,6S)	ΔIIIs = ΔHexUA(2S) − GlcN(NS,3S)
ΔIs = ΔHexUA(2S) − GlcN(NS,3S,6S)	IVs_gal_ = GalA − GlcNS
IIs_gal_ = GalA − GlcN(NS,6S)	IIs_epoxy_ = GulA2,3epo − GlcN(NS,6S)
IIs_glu_ = GlcA − GlcN(NS,6S)	IIIs_id_ = IdoA(2S) − GlcNS
IVs_glu_ = GlcA − GlcNS	Is_id_ = IdoA(2S) − GlcN(NS,6S)
IIs_glu_ = GlcA − GlcN(NS,3S,6S)	IVs_glu_ = GlcA − GlcN(NS,3S)
Glyserox = Oxidized glycoserine (ΔGlcA-Gal-Gal-Xyl-COOH)	
ΔU(x,y,z) = Δ-unsaturated oligosaccharide, x saccharides units, y sulfates, z *N*-acetyl	
ΔU(x,y,z)^sulf^ = ΔU(x,y,z) with sulfanilic acid reductive amination	
G(x,y,z) = Oligosaccharide with a glucosamine at its non-reducing end, x saccharides units, y sulfates, z *N*-acetyl	
G(x,y,z)^sulf^ = G(x,y,z) with sulfanilic acid reductive amination	
Mw 595^sulf^ = Oligosaccharide at Mw 595 Da with sulfanilic reductive amination (595 + 157 Da)	
The iduronic (id) or glucuronic (glu) structure of uronic acids is indicated for oligosaccharides, e.g., ΔIs-III_id_ Underlined disaccharides have a 3-*O* sulfated glucosamine, e.g., IIs_glu_ (GlcA-GlcNS,3S,6S)	

Underlined disaccharides have a 3-O sulfated glucosamine.

**Table 2 molecules-25-05553-t002:** Quantification of building blocks (% *w*/*w*) for PMH, OMH, and BMH (NAc: % *N*-acetylated glucosamines; 6-OH: % 6-OH glucosamines; 2-OH: % 2-OH uronic acids; 3-*O*S: % 3-*O* sulfated glucosamines).

	Heparin	PMH	OMH	BMH
**Unsaturated Building Blocks**	ΔHexUA-GlcNAc	2.3	1.6	3.0
	ΔHexGalA-GlcNS	0.0	0.0	0.2
	ΔHexUA-GlcNS	2.3	1.1	2.9
	ΔHexUA-GlcNAc(6S)	2.3	1.4	0.7
	ΔHexUA(2S)-GlcNAc	1.4	0.5	1.0
	ΔHexGalA-GlcN(NS,6S)	0.2	0.2	0.2
	ΔHexUA-GlcN(NS,6S)	8.6	9.0	6.9
	ΔHexUA(2S)-GlcN(NS)	6.0	5.6	25.2
	ΔHexUA(2S)-GlcNAc(6S)	1.4	0.6	0.2
	ΔHexUA-GlcNAc(6S)-GlcA-GlcN(NS,3S)	0.9	0.5	0.9
	ΔHexUA-GlcN(NS,3S,6S)	0.1	0.3	0.2
	ΔHexUA(2S)-GlcN(NS,3S)	0.1	0.1	1.3
	ΔHexUA(2S)-GlcN(NS,6S)	61.8	65.6	43.9
	ΔHexUA-GlcNAc(6S)-GlcA-GlcN(NS,3S,6S)	3.7	3.4	1.1
	ΔHexUA(2S)-GlcN(NS,3S,6S)	0.3	0.5	1.6
	ΔHexUA-GlcN(NS, 6S)-GlcA-GlcN(NS,3S,6S)	0.2	0.8	0.4
	ΔHexUA(2S)-GlcNAc(6S)-GlcA-GlcN(NS,3S,6S)	0.2	0.2	0.1
	ΔHexUA(2S)-GlcN(2S,6S)-GlcA-GlcN(NS,3S,6S)	0.4	0.9	0.6
**NRE Building Blocks**	GlcNS	0.1	0.4	0.2
	GlcNAc(6S)	0.1	0.0	0.0
	GlcN(NS,3S)	0.1	0.2	0.1
	GlcN(NS,6S)	0.5	0.5	0.5
	U(2,2,0)	0.5	0.2	0.7
	GlcA-GlcN(NS,3S,6S)	0.9	<0.1	<0.1
	IdoA(2S)-GlcN(NS,6S)	0.7	0.6	2.1
	GlcN(NS,3S,6S)	0.3	1.1	0.8
	G(3,4,0)	0.2	0.5	N.D.
	G(3,5,0)	0.2	0.3	0.2
**NRE**	% NRE (Monosaccharides)	2.0	2.3	2.5
	% Glucosamines	51.2	81.6	48.7
	% uronic acids	48.8	18.4	51.3
	Is_id_/IIs_glu_	0.7	7.7	36.4
	% 3-0 Sulfation	36.4	37.7	23.7
**Heparin**	**Sulfates/Carboxylates**	**2.48**	**2.63**	**2.31**
	NAc	12.6	8.0	7.8
	6-OH	16.3	12.6	38.7
	2-OH	26.4	22.3	20.2
	3-*O*S	5.0	5.7	5.4

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
