# Peer review of "Specific Non-Reducing Ends in Heparins from Different Animal Origins: Building Blocks Analysis Using Reductive Amination Tagging by Sulfanilic Acid"

_molecules, 2020, doi:10.3390/molecules25235553_

Round 1

Reviewer 1 Report

The aim of this study is to improve techniques of characterizing the non-reducing end (NRE) of low-molecular-weight heparins (LMWHs). Special emphasis is given to enoxaparin, generated through ß-eliminative cleavage of so-called unfractionated heparin. Novel methods are developed for separation and MS-based characterization of products obtained upon exhaustive digestion of LMWH with bacterial heparinases, in particular following derivatisation through reductive amination with sulfanilic acid. A special ambition of the study is to compare NREs of LMWHs derived from heparin starting materials obtained from various animal sources; this aspect would seem primarily dictated by commercial concerns, of limited interest to the average reader.

However, some of the findings are of more general significance and deserve emphasis. The study identifies two predominant types of NREs, one terminating in glucosamine, the other in saturated hexuronic acid residues. The former structure is explained by the mechanism believed to generate unfractionated heparin, through herparanase (an endo-ß-glucuronidase) cleavage of the extended chains of the serglycin proteoglycan. Non-reducing-terminal hexuronic acid residues, on the other hand, are not readily rationalized in terms of currently recognized cellular mechanisms, nor processes of LMWH manufacture. The suggestion that these structures represent termini of the initial, proteoglycan-attached chains could conceivably be approached by NRE identification, using the methods developed, of such chains. Another striking finding is the high proportion of 3-O-sulfated NRE structures, possibly representing remnants of antithrombin-binding sequences. Then why would a proportion of such sequences remain intact, as required to explain the anticoagulant activity of heparin products?

Unfortunately, these issues are obscured by the massive amount of information relating to methodology. These dominant parts of the ms. appear inaccessible not only because of the number of complex chromatograms, but also due to the abundance of innovative structural symbols used throughout the text and the many figures. How many readers will manage to digest this material? Even if the information is prerequisite to the interesting conclusions referred to above, it should be presented in a less discouraging format – maybe in the supplementary section? But then this section is already overloaded with material.

Some issues of relevance to the presentation could be commented upon. How are GlcA and IdoA in the intact polymer distinguished after heparinase cleavage of adjacent glucosaminidic bonds? How explain the GlcNAc-IdoA-GlcNS trisaccharide structure in Fig. 4, when C5-epimerization of GlcA is precluded by a GlcNAc nonreducing neighbor?

Author Response

RESPONSE TO REVIEWER 1

A special ambition of the study is to compare NREs of LMWHs derived from heparin starting materials obtained from various animal sources; this aspect would seem primarily dictated by commercial concerns, of limited interest to the average reader.

However, some of the findings are of more general significance and deserve emphasis. The study identifies two predominant types of NREs, one terminating in glucosamine, the other in saturated hexuronic acid residues. The former structure is explained by the mechanism believed to generate unfractionated heparin, through herparanase (an endo-ß-glucuronidase) cleavage of the extended chains of the serglycin proteoglycan. Non-reducing-terminal hexuronic acid residues, on the other hand, are not readily rationalized in terms of currently recognized cellular mechanisms, nor processes of LMWH manufacture. The suggestion that these structures represent termini of the initial, proteoglycan-attached chains could conceivably be approached by NRE identification, using the methods developed, of such chains. Another striking finding is the high proportion of 3-O-sulfated NRE structures, possibly representing remnants of antithrombin-binding sequences. Then why would a proportion of such sequences remain intact, as required to explain the anticoagulant activity of heparin products?

Unfortunately, these issues are obscured by the massive amount of information relating to methodology. These dominant parts of the ms. appear inaccessible not only because of the number of complex chromatograms, but also due to the abundance of innovative structural symbols used throughout the text and the many figures. How many readers will manage to digest this material? Even if the information is prerequisite to the interesting conclusions referred to above, it should be presented in a less discouraging format – maybe in the supplementary section? But then this section is already overloaded with material.

We thank the reviewer for his discussion and comments on our manuscript. The method we developed indeed has given insight to an aspect of heparin which was little studied and has in turn has raised numerous interesting questions around the biological processes leading to production of commercial heparins. However, our initial objective was not the characterization of these differences but instead the problem of differentiation in LMWH and in heparins. For drug companies and agencies, this problem is of the utmost concern with significant consequences for patient health. The 2008 heparin crisis, due to contamination with persulfated chondroitin sulfate, resulted in around 150 death in the US alone. Thus, our purpose was originally to develop a new analytical tool to differentiate heparin sources.

We are aware that the manuscript is quite complex with a large volume of methodological information. However, given that the focus was to communicate a novel technique, we feel that we are required to include the level of information presented in order to provide a comprehensive resource for other researchers seeking to replicate our results or apply them to their own research. We have retained as much data as possible in the supplementary information, and this is supported by reviewer 3 who mentioned that the supplementary information greatly enhanced understanding of this work.  

We understand that our structural symbols, quite numerous in the text, are tedious for the reader. Following the advice of the reviewers, we have included a nomenclature table within the text for easy reference, and all structural symbols were defined in the text at first appearance and in figure legends. 

Some issues of relevance to the presentation could be commented upon. How are GlcA and IdoA in the intact polymer distinguished after heparinase cleavage of adjacent glucosaminidic bonds?

Heparinase cleavage functions through a b elimination that destroys the configuration of the uronic acid, making it impossible to distinguish between GlcA and IdoA. The only way to identify the configuration of the uronic acid is through nitrous depolymerization to retain the configuration but prevent the generation of any UV absorbing moiety. While very interesting, this topic is beyond the scope of our manuscript.

 How explain the GlcNAc-IdoA-GlcNS trisaccharide structure in Fig. 4, when C5-epimerization of GlcA is precluded by a GlcNAc nonreducing neighbor 

The reviewer is perfectly right. The iduronic configuration on the scheme was misleading and we corrected it. In fact, these trisaccharides were identified by LC/MS with no information on the configuration of the uronic acid. The structure GlcNAc-IdoA-GlcNS, corresponding to 718 Da was the one which is present in smallest amount compared to the 2 others at 798 Da and 878 Da. Two trisaccharides at 798 and 878 Da were obtained by the action of D Glycuronidase on sulfanilic tagged digested heparin with corresponding structures GlcNAc(6S)-GlcA-GlcN(NS,3S) and GlcNAc(6S)-GlcA-GlcN(NS,3S,6S), both with a glucuronic acid configuration, in line with the C-5 epimerization rule. By contrast, as mentioned in the discussion in chapter 2.5, we isolated and fully identified a trisaccharide GlcNAc-IdoA(2S)-GlcN(NS,6S) which apparently opposed it. So, the C-5 epimerization rule, could just reflect a general behavior with possible exceptions. 

Reviewer 2 Report

   The authors developed the new analytical method of non-reducing end of heparin by heparinase digestion and strong anion-exchange chromatography. In addition, they identified that the novel heparin oligosaccharides at non-reducing terminal, and demonstrated the difference in the structures among heparin sources. Furthermore, the methodology is well designed, the manuscript is well written, and the data agree with the conclusions. However, a few minor concerns should be addressed.

Thank you very much for allowing me to consider the work.

Sincerely,

Major Comment #1

   The authors describe the abbreviations, nomenclature, and symbols of heparin structures in pages 18-19, lines 515-516. However, these abbreviations and symbols should be described in the respective first appearance of main text. Furthermore, the structural symbols of heparin were very complicated for the audiences. Thus, the structures of di- and/or oligo-saccharides from heparin as well as heparan sulfate should be described each sugar abbreviation in the main text. For example, the sentence “The ratio of absorbances at 265 nm for ∆Issulf and at 232 nm for ∆Is at pH 3 is 2.5” in page 3, line 128, is corrected into “The ratio of absorbances at 265 nm for ∆HexUA(2S)-GlcN(NS,6S)-4ABS or -SA (∆Issulf) and at 232 nm for ∆HexUA(2S)-GlcN(NS,6S) (∆Is) at pH 3 is 2.5”. Namely, the structures or sequences di- and oligo-saccharides from heparin are described together with the respective abbreviation in the main text. In the Figures and Tables, the structures/sequences, such as ∆HexUA(2S)-GlcN(NS,6S) (∆Is), should be described in the legend (Figs. 1-3, 5-9, Table 1, Supplementary Figs and Tables).

In addition, the author described that the underline indicates 3-O-sulfate, which is also complicated for the audiences. The reviewer did not recommend such abbreviation.

Major Comment #2

   The reviewer felt that the scheme of the analysis may be necessary for understanding the methodology.

Minor Comment #3

   The several peaks of di- and oligo-saccharides on chromatograms overlaied onto the respective peak from the other chromatograms. Thus, the peaks from heparin do not overlay onto the respective peak from other chromatograms.

Minor Comment #4

   The author may discuss the structures of non-reducing end from heparin and/or heparan sulfate, which are yielded by treatment with nitrous acid.

Other Minor Comments:

1) The Figure “a” and “b” should be described upper left or right side without circle in each figure.

2) A couple of character “beta” and “delta” were unreadable character in lines 258, 260, 350, 500)

3) Figure 5 may be deleted or moved to Supplemental Fig.

4) Supplementary Figs. 1S-23S in the main text as well as the legends ––> Supplementary Figs. S1-S23

5) Supplementary Tables 1S-10S in the main text as well as the legends ––> Supplementary Tables S1-S10

6) p2, line 67: hyaluronic acid ––> hyaluronan

7) p2, line 91: Low molecular weight heparins ––> LMWHs

8) p5, line 180; p6, line 188; p9, line 251; p10, line 270; p11, line 290: The “black line” and “minus” were confused.

9) p7, Fig. 4: The connection of monosaccharide, -O-, at the non-reducing end in trisaccharides was linear, but the others were round. In addition, the “Mw” should be deleted and describe the only a “Mw” in each line.

10) p10 and 11, Figs 7, and 8: heparin digest ––> heparinase digest

11) p 11, Figs 8: Glycuronidase ––> ∆Glycuronidase

   Glc NS, 6S-IISglusulf  ––> without space between Glc and NS

12) p11, line 298: (33) ––> [33]

13) p12, line 314: Each black line should be corrected to the respective colored line.

14) p13, lines 347, 350: “[34)” and “[15)”  ––>  “[34]” and “[15]”

15) p14, Table 1: The di- and oligo-saccharides should be described the respective sequence but not the symbols. Ex) ∆IVa ––> ∆HexUA-GlcNAc

16) p15, line 379: Glucosamine ––> glucosamine

17) p15, lines 393, 404, 405: sulfatation ––> sulfation

18) p15, lines 405, 406, 416: Glc(NS, 3S, 6S) ––> GlcN(NS, 3S, 6S)

19) p16, line 424: More details of the beta-D-glucuronidase should be described such as from bovine liver, type B.

20) p18: ∆GlcA ––> ∆UA

21) p18: ∆Va ––> ∆Iva ?

Author Response

RESPONSE TO REVIEWER 2

Major Comment #1

The authors describe the abbreviations, nomenclature, and symbols of heparin structures in pages 18-19, lines 515-516. However, these abbreviations and symbols should be described in the respective first appearance of main text. Furthermore, the structural symbols of heparin were very complicated for the audiences. Thus, the structures of di- and/or oligo-saccharides from heparin as well as heparan sulfate should be described each sugar abbreviation in the main text. For example, the sentence “The ratio of absorbances at 265 nm for ∆Issulf and at 232 nm for ∆Is at pH 3 is 2.5” in page 3, line 128, is corrected into “The ratio of absorbances at 265 nm for ∆HexUA(2S)-GlcN(NS,6S)-4ABS or -SA (∆Issulf) and at 232 nm for ∆HexUA(2S)-GlcN(NS,6S) (∆Is) at pH 3 is 2.5”. Namely, the structures or sequences di- and oligo-saccharides from heparin are described together with the respective abbreviation in the main text. In the Figures and Tables, the structures/sequences, such as ∆HexUA(2S)-GlcN(NS,6S) (∆Is), should be described in the legend (Figs. 1-3, 5-9, Table 1, Supplementary Figs and Tables).

We thank this reviewer for his feedback and for their precise correction of our manuscript. We appreciate that the manuscript is heavy with symbols; unfortunately, due to the analytical nature of the study and the number of new building blocks identified it is difficult to reduce further. We have included Table 1 describing nomenclature and symbols at the beginning of the manuscript for easy reference and have explained each symbol at first appearance. Symbols used in figures were also fully described in the legends

In addition, the author described that the underline indicates 3-O-sulfate, which is also complicated for the audiences. The reviewer did not recommend such abbreviation

We respectfully ask to keep this symbol. We are aware that it is not a frequently utilized nomenclature, however, given the already extensive structural information presented in the text we sought to reduce this as much as possible, for which this is helpful. Additionally, this abbreviation has been used in other publications and is not unique to ours, for example – Franco Spelta, Lino Liverani, Alessandra Peluso, Maria Marinozzi, Elena Urso, Marco Guerrini and Annamaria Naggi, Orthogonal Methods for Characterizing Heparin Batches Composition, Frontiers in Medicine April 2019 | Volume 6 | Article 78. Moreover, this notation style was accepted in our previous article published in Molecules, titled ‘New insights in thrombin inhibition structure–activity relationships by characterization of octadecasaccharides from Low Molecular Weight Heparin’, Molecules 2017, 22, 428.  

Major Comment #2

   The reviewer felt that the scheme of the analysis may be necessary for understanding the methodology.

We agree with the reviewer’s comment and have added a schematic (Figure 3).

Minor Comment #3

   The several peaks of di- and oligo-saccharides on chromatograms overlaied onto the respective peak from the other chromatograms. Thus, the peaks from heparin do not overlay onto the respective peak from other chromatograms.

Unfortunately, we did not understand the reviewer’s comment here and so no change has been implemented to the manuscript. We apologize.

Minor Comment #4

   The author may discuss the structures of non-reducing end from heparin and/or heparan sulfate, which are yielded by treatment with nitrous acid.

We did not talk about Dalteparin and Fraxiparin because we do not have any real experience of the specific building blocks characteristic of LMWH obtained by nitrous depolymerization. We only analyzed one sample of Dalteparin (a generic batch from China) and while we could quite logically detect major NREs of oligosaccharide chains (IdoA(2S)-GlcNS(NS,6S)-SA and IdoA(2S)-GlcNS(NS)-SA) as expected, we could not explain a peak doubling for IdoA(2S)-GlcNS(NS,6S)-SA which was not observed on heparin NRE. The tetrasaccharide DUA(2S)-GlcN(NS,6S)-Mnt6S  was also detected. Other mannose derivatives were detected but all building blocks were not clearly identifiable. Moreover, we felt that the article was already long, and that it would be not interesting to lengthen it with data on a LMWH with which we are not familiar. We hope this is acceptable.

Other Minor Comments

1) The Figure “a” and “b” should be described upper left or right side without circle in each figure.

The modification is done

2) A couple of character “beta” and “delta” were unreadable character in lines 258, 260, 350, 500)

The corrections were done

3) Figure 5 may be deleted or moved to Supplemental Fig.

The figure was deleted.

4) Supplementary Figs. 1S-23S in the main text as well as the legends ––> Supplementary Figs. S1-S23

5) Supplementary Tables 1S-10S in the main text as well as the legends ––> Supplementary Tables S1-S10

All references to supplementary data were modified

6) p2, line 67: hyaluronic acid ––> hyaluronan

7) p2, line 91: Low molecular weight heparins ––> LMWHs

Both modifications were done

8) p5, line 180; p6, line 188; p9, line 251; p10, line 270; p11, line 290: The “black line” and “minus” were confused.

Colored lines have been enhanced to ensure easier visual distinction from minus signs or other punctuation.

9) p7, Fig. 4: The connection of monosaccharide, -O-, at the non-reducing end in trisaccharides was linear, but the others were round. In addition, the “Mw” should be deleted and describe the only a “Mw” in each line.

The figure has been modified.

10) p10 and 11, Figs 7, and 8: heparin digest ––> heparinase digest

11) p 11, Figs 8: Glycuronidase ––> ∆Glycuronidase

   Glc NS, 6S-IISglusulf  ––> without space between Glc and NS

Figures 7 and 8 were modified. When we use “heparin digest”, we did not talk about heparinase digest but digested heparin. It was thus replaced by Sulfanilic heparin digest

12) p11, line 298: (33) ––> [33]

Corrected.

13) p12, line 314: Each black line should be corrected to the respective colored line.

This has been corrected.

14) p13, lines 347, 350: “[34)” and “[15)” ––> “[34]” and “[15]”

The modifications were done

15) p14, Table 1: The di- and oligo-saccharides should be described the respective sequence but not the symbols. Ex) ∆IVa ––> ∆HexUA-GlcNAc

Symbols were replaced by the whole sequence on the Table

16) p15, line 379: Glucosamine ––> glucosamine

Corrected.

17) p15, lines 393, 404, 405: sulfatation ––> sulfation

Corrected.

18) p15, lines 405, 406, 416: Glc(NS, 3S, 6S) ––> GlcN(NS, 3S, 6S)

Corrected.

19) p16, line 424: More details of the beta-D-glucuronidase should be described such as from bovine liver, type B.

Corrected, the b-glucuronidase used was indeed from bovine liver, type B.

20) p18: ∆GlcA ––> ∆UA

Corrected.

21) p18: ∆Va ––> ∆Iva ?

Corrected.

Reviewer 3 Report

The paper by Mourier describes the development of a modification to an existing method within their laboratory that allows for ‘species’ identification of heparin. This method is an extension of the technique of disaccharide analysis which can be used for heparin assessment but is a required technique for the preparation of enoxaparin. The addition of reductive amination, effecting the non-reducing ends, and LC/MS yields a method that can be used to identify unique building blocks in heparinase digested heparin samples. The ability to determine differences in the proportion of these building blocks in heparin from different animal sources was presented and will be of interest to the field of heparin analysis.

The paper presents a clear methodology and rationale for the development steps taken, with supporting data within the paper but also in the extensive supplementary data. Overall, the paper is well written, with the aims, methods and results well-presented and clear.  The discussion and conclusion nicely match the aims/objectives of the work presented with distinctions between bovine, ovine and porcine heparin building blocks being clearly determined.  It would be interesting to see the sensitivity of this method for determining the presence of cross species contamination in heparin, but it is clear this was outside the scope of this methodology paper. 

However, as a minor point there are several occurrences where connections within the paper could be improved. One observation is that there is reference to LMWH, namely enoxaparin, prepared from different sources of heparin within the introduction (line 94) as part of the development process. Within the discussion, line 372 to 381, reference is only made to an ovine prepared material with no mention of an enoxaparin from bovine heparin, despite an indication in line 363. Perhaps this is an oversight, given that the author has indicated the difficulties of studying enoxaparin with this technique due to the presence of 1-6 anhydro linkage, hence semuloparin was used for further development work as indicated in section 2.3 - it is unclear if this was semuloparin from alternative heparins. A second observation is the cross referencing to the quite extensive supplementary data whose presence greatly enhances understanding of the steps and analysis of this work. The issue is that not all the figures and tables within the supplementary data are referred to in the main text. As these figures/tables, in this reviewer opinion, greatly enhance understanding of this work, it would be beneficial for the cross referencing to be more complete.

Author Response

RESPONSE TO REVIEWER 3

It would be interesting to see the sensitivity of this method for determining the presence of cross species contamination in heparin, but it is clear this was outside the scope of this methodology paper.

It was indeed out of the scope of this paper; however, it is an interesting consideration. The choice of sulfanilic acid was not dictated by the sensitivity of the detection but by the compatibility with methods used for the separation of sulfated oligosaccharides, particularly Gel Permeation Chromatography and anion exchange chromatography. When we determined the response coefficient between 265 nm (the maximum absorbance of oligosaccharides tagged with sulfanilic acid), and 232 nm (that of unsaturated oligosaccharides), we found only 2.5, whereas we expected higher. The interferences at 265 nm are less abundant than at 232 nm. Moreover, compared to the classical building block methods on silica SAX, the gain in peak efficiency is quite significant, so that the sensitivity gain with the new method should reach at least a factor 4 or 5.

However, as a minor point there are several occurrences where connections within the paper could be improved. One observation is that there is reference to LMWH, namely enoxaparin, prepared from different sources of heparin within the introduction (line 94) as part of the development process. Within the discussion, line 372 to 381, reference is only made to an ovine prepared material with no mention of an enoxaparin from bovine heparin, despite an indication in line 363. Perhaps this is an oversight, given that the author has indicated the difficulties of studying enoxaparin with this technique due to the presence of 1-6 anhydro linkage, hence semuloparin was used for further development work as indicated in section 2.3 - it is unclear if this was semuloparin from alternative heparins.

Thank you for this feedback, we have altered the manuscript to more accurately answer these points. Firstly, the choice of semuloparin against enoxaparin for the separation of tetrasaccharidic building blocks was due to many interesting factors. Semuloparin is a LMWH which, like enoxaparin, is obtained from porcine heparin, but is much smaller (2500 Da compared to 4500 Da, respectively), and thus, the tetrasaccharide fraction is much more abundant than in enoxaparin. But the factor which was the most interesting for us was the simplicity of the fraction. In enoxaparin, alkaline fingerprints, such as 1.6 anhydro derivatives, odd oligosaccharides, and mannosamine epimerization, results in complex fractions, even the smallest like tetrasaccharides. Semuloparin has the considerable advantage of containing very few alkaline fingerprints, just some mannosamine epimers but in much smaller amount than in enoxaparin.

The reference in our manuscript to the study of the LMWH generated by other heparin sources, OMH and BMH, was done because the comparison of the ovine LMWH with enoxaparin was for us the opportunity to discover the different NREs between OMH and PMH. OMH and PMH have structural differences, such as the percentage of acetylated glucosamines, but these differences can only be detected by statistics and, until now, no specific marker had ever been identified. In the comparison of ovine LMWH with enoxaparin, you can also detect in fractions fully compatible with analysis such as tetrasaccharides, structural features that cannot be detected in an unfractionated heparin. In the ovine LMWH, we detected the increased amount of NRE glucosamine and the absence of NRE uronic acid, always observed on all LMWH fractions from porcine origin previously analyzed. The need to measure the origin of the phenomenon, in the heparin batch, resulted in the development of the described method. For the LMWH generated by BMH, the incredible number of differences observed with enoxaparin, paradoxically hid the influence of the heparin NRE.

A second observation is the cross referencing to the quite extensive supplementary data whose presence greatly enhances understanding of the steps and analysis of this work. The issue is that not all the figures and tables within the supplementary data are referred to in the main text. As these figures/tables, in this reviewer opinion, greatly enhance understanding of this work, it would be beneficial for the cross referencing to be more complete.

We thank the reviewer for his observation. All to supplementary figures or tables have been checked and included where required.

Round 2

Reviewer 1 Report

The authors have chosen not to further consider the issues of more general biological interest raised in my previous review. The accessibility of the manuscript has been improved by revision, but remains problematic. As it stands, I would consider the report of interest to LMWH manufacturers but hardly to a wider readership. I am skeptical to the statement (in the rebuttal) that the results of the study would be of importance to patient health.